

# Synergism between calcium nitrate applications and fungal endophytes to increase sugar concentration in *Festuca sinensis* under cold stress

Lianyu Zhou[1,2], Chunjie Li[1], James F. White[3] and Richard D. Johnson[4]

[1] State Key Laboratory of Grassland Agro-ecosystems; Key Laboratory of Grassland Livestock Industry Innovation, Ministry of Agriculture and Rural Affairs; Engineering Research Center of Grassland Industry, Ministry of Education; Gansu Tech Innovation Centre, Lanzhou University, Lanzhou, Gansu, China
[2] Key Laboratory of Medicinal Plant and Animal Resources of the Qinghai-Tibetan Plateau, School of Life Science, Qinghai Normal University, Xining, Qinghai, China
[3] Department of Plant Biology, Rutgers University, New Brunswick, NJ, United States of America
[4] Grasslands Research Centre, AgResearch Limited, Palmerston North, New Zealand

Corresponding author
Chunjie Li, chunjie@lzu.edu.cn

## ABSTRACT

*Epichloë* endophytes have been shown to increase tolerance to biotic and abiotic stresses in many cool-season grasses. We investigated the impact of endophyte infection of *Festuca sinensis*, on root metabolic activity, photosynthetic pigments, leaf relative water content (RWC) and soluble carbohydrates in a field experiment carried out during chilling and irrigation with $Ca(NO_3)_2$. A highly significant ($P < 0.001$) correlation for *Epichloë* endophytes was observed for root metabolic activity. $Ca(NO_3)_2$ affected very significantly root metabolic activity and total chlorophyll ($P < 0.001$). Low temperature led to highly significant ($P < 0.001$) reductions in root metabolic activity, RWC, total chlorophyll, chlorophyll a/b ratio, and carotenoid contents. In addition, the fructose concentrations of shoots were greater on the 14th day than on the 28th day and before treatment, whilst the glucose concentration of roots was much higher on the 28th day than before and after 14 days treatment. Moreover, our results indicated that the addition of calcium nitrate contributed to higher levels of total chlorophylls, soluble sugars, sucrose, fructose or glucose in the shoots and roots in both E+ and E- plants during long periods of chilling. These results suggest that *Epichloë* endophyte infection and/or exogenous calcium nitrate can confer better tolerance to cold stress.

## INTRODUCTION

Low temperature is one of the main abiotic stresses that retards growth and development of many plants (*Nayyar, Bains & Kumar, 2005*). There is considerable evidence that exposure to low temperature causes a variety of morphological, physiological and biochemical responses, including declines in chlorophyll content (*Hansen, Vogg & Beck, 1996; Lootens, Van Waes & Carlier, 2004*), root metabolic activity (*Sun et al., 2016*), leaf relative water content (RWC) (*Burchett, Niven & Fuller, 2006*), and accumulation of soluble

carbohydrates (*Klotke et al., 2004*; *Stupnikova et al., 2002*). Nitrogen uptake is significantly influenced by low temperature and lack of growth activity of the plant (*Dong et al., 2001*).

Calcium is a critical nutrient that plays a significant role in maintaining the structure and function of the cell wall and cell membrane. Many studies have demonstrated that calcium regulates the processes of plant growth and development, and acts as a second signal to regulate responses of plants to stresses (*Bush, 1995*; *Bowler & Fluhr, 2000*; *Hepler, 2005*), such as drought (*Xu, Li & Zhang, 2013*), temperature (*Monroy, Sarhan & Dhindsa, 1993*; *Tan et al., 2011*; *Yang et al., 2013*; *Karpets et al., 2016*; *Yang et al., 2016*), oxidate (*Schmitz-Eiberger, Haefs & Noga, 2002*), salt (*Navarro, Martínez & Carvajal, 2000*) and metals (*Min et al., 2012*; *Rahman et al., 2016*). Exogenous calcium has been demonstrated to confer protection against cold stress by alleviating chlorophyll and carotenoid degradation (*Schaberg et al., 2011*; *Feng et al., 2010*), increasing antioxidant enzyme activity, increasing root metabolic activity (*Li, Gao & Liu, 2011*; *Feng et al., 2010*; *Liu et al., 2012*), and enhancing sugar contents (*Halman et al., 2008*; *Hawley et al., 2006*). Cold stress promotes endogenous $Ca^{2+}$ level in leaves and roots, moreover, exogenous application of $Ca^{2+}$ increased endogenous $Ca^{2+}$ content under control and cold conditions (*Shi et al., 2014*).

Plant adaptation to stress may be mediated by association with microorganisms (*Rodriguez & Redman, 2008*; *Redman et al., 2011*; *Yin et al., 2014*). *Epichloë* endophytes (formerly *Neotyphodium* spp.; *Leuchtmann et al., 2014*) have been found in many cool-season grasses and can improve the ability of host grasses to withstand biotic and abiotic stresses (*Johnson et al., 2013*; *Yin et al., 2014*; *Xia et al., 2016*; *Ma, Christensen & Nan, 2015*; *Song et al., 2015*). For example, grass infection with *Epichloë* endophytes improves photosynthetic pigment abundance (*Rozpadek et al., 2015*), and confers benefits to grasses in response to cold stress including increased root biomass and increased fungal secondary metabolite production (*Latch, Hunt & Musgrave, 1985*; *Zhou et al., 2015a*; *Chen et al., 2016*). In addition, other symbioses are reported to affect photosynthetic pigment degradation, leaf relative water content, soluble sugars and other physiological markers for chilling stress (*Mishra et al., 2011*; *Zhou et al., 2012*; *Ding et al., 2011*; *Wang et al., 2016*). In such instances, exogenous chemical applications or symbiotic technology serve to improve plant adaptation to low temperature stress (*Halman et al., 2008*; *Yang et al., 2016*; *Jeon et al., 2010*; *Zhou & Guo, 2009*; *Redman et al., 2011*). The combined application of microorganism and plant growth regulator has been found to protect plant from low temperature stress (*Zhou et al., 2012*). However, information about the influence of exogenous calcium nitrate and *Epichloë* on physiological response of plant adaptation to cold stress is rather scanty.

*Festuca sinensis* is an important cool-season grass species grazed by cattle and sheep, especially in cool and semi-arid regions of China (*Nan & Li, 2000*). This grass species is frequently host to an asexual symptomless *Epichloë* sp. that resides in the apoplastic spaces of the leaf sheaths, culms and seeds (*Peng et al., 2013*). Several reports have indicated that *Epichloë* endophytes improve the response of *F. sinensis* to stresses such as drought, pathogens and cold under controlled conditions (*Peng et al., 2013*; *Zhou et al., 2015b*). Cold winter conditions can lead to damage or death of grasses. However, despite this survival of

*F. sinensis* it is still stressed under long-term cold exposure in grasslands of northwestern China.

A previous study has suggested that cold stress negatively affects nitrogen uptake by plants (*Dong et al., 2001*). The aim of the present study was to examine the role of exogenously applied calcium nitrate and *Epichloë* endophytes in stimulation of root metabolic activity, photosynthetic pigment accumulation, RWC and soluble carbohydrates, and how these may improve *F. sinensis* survival under cold conditions in the field.

# MATERIALS & METHODS

## Growth

*Festuca sinensis* seeds (E+) that were naturally infected with the endophyte were collected in September 2012 from Xiahe county, Gansu Provence in China (3,000 m, 98°23′E, 34°23′N). To obtain endophyte free (E- ) seeds, *Epichloë* endophyte infected seeds (E+) were treated with 300 times dilution of fungicide thiophanate-methyl for 8 h to kill the fungus (*Yao et al., 2013*). The infection status of each plant was determined via microscopic (Olympus, Japan) examination of aniline blue-stained leaf sheath pieces. Vegetatively-propagated E+ and E- plants, grown in a constant temperature greenhouse (soil:nutritive medium, 1:1, v/v, 22 °C, 16 h light/8 h dark) as previously described (*Zhou et al., 2015a*) were used in subsequent experiments. The clones consisting of one tiller were planted during May 2014 in the experimental field of the College of Pastoral Agriculture Science and Technology, Yuzhong campus of Lanzhou University (loess soil, 104°39′E, 35°89′N, Altitude 1,653 m), and were watered as required. Before the onset of experimental treatments, the soil was tilled to reduce differences in soil fertility. The trials were arranged using a split-split-plot design with a total of 216 plants of *F. sinensis*, where three $Ca(NO_3)_2$ concentrations were randomly assigned into three replicates as main plot treatments, two endophyte statuses (i.e., E+ and E- ) were randomly assigned into the main plots as split plot treatments, and randomly selected plants were sampled at three times in two strips (two plants each time) as the split-split plots in two strips (Fig. S1). There were 18 treatments (2 endophyte types ×3 $Ca(NO_3)_2$ concentrations ×3 sampling times), with three repetitions for each treatment and 12 plants for each repetition (i.e., 36 plants for each treatment). There was a distance of 22 cm between strips (i.e., endophyte types) and 18 cm between plants within a strip. For split plot, plot size was 20 m$^2$ with two surrounding-protection strips and two sampling strips. The strips of plants were 0.5 m apart from each other. Plants were grown for 5 months under natural conditions. After this period, plants were irrigated from 29 October to 28 November once a week for 4 weeks with 15 mL $Ca(NO_3)_2$ solution at three concentrations, 0 (CK), 25 and 50 mM, respectively. The herbage and roots of 6 different plants under each $Ca(NO_3)_2$ solution condition for either E+ or E- plants were randomly harvested from three repetitions in the morning on day 0 (before plant treatment), 14 and 28, respectively. Daily air temperature ranged from 3 °C to 14 °C during the day and −7 ° C to 2 °C during the night during the experiment according to the data from http://www.tianqihoubao.com/lishi/yuzhong.html (Fig. 1).
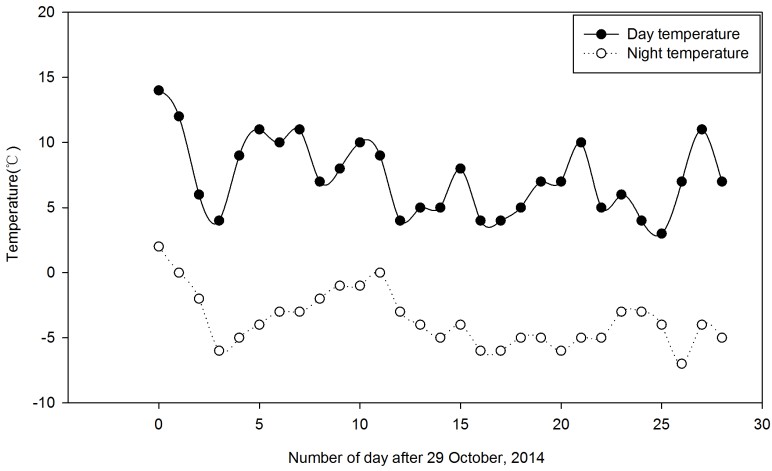

**Figure 1  Daily air temperature profile of the field during the study period (29 October to 28 November, 2014).** Each data point indicates the average temperature.

## Determination of root metabolic activity

Root metabolic activity was measured using the modified *2, 3, 5-* triphenyltetrazolium chloride (TTC) method (*Li, 2000*). This method provides an estimate of metabolic activity in roots by reflecting aerobic respiration rates (*Li, 2000*). To conduct this assay, root tips (0.5 g fresh weigh, FW) were dipped in a solution containing five mL of 0.4% TTC and five mL of 1/15 M phosphate buffer solution (pH 7.0), and incubated at 37 °C for 1 h in the dark. Then, 2 mL of 1 M $H_2SO_4$ was added to the mixture to stop the reaction. The root was removed carefully, wiped and then ground with a small amount of ethyl acetate and quartz sand in a mortar to extract triphenyl formazan (TTF). The residue was washed 2–3 times with a small amount of ethyl acetate and added to the first extraction and made up to a volume of 10 mL. Root metabolic activity was measured using a spectrophotometer (SP-723) with an absorbance of 485 nm and using $H_2SO_4$ as a blank.

## Analysis of chlorophyll and carotenoid contents

Chlorophyll and carotenoid contents were measured according to the modified method of *Li (2000)*. Leaves (0.05 g FW) were cut and extracted with 80% acetone for 48 h in the dark. Absorbance was quantified at 663, 645 and 470 nm using a spectrophotometer. Chlorophyll a, chlorophyll b, total chlorophyll and carotenoid contents were calculated according to the following equations: chlorophyll a ($C_a$) = (12. 72 × $A_{663}$) (2.59 × $A_{645}$), chlorophyll b ($C_b$) = (22.88 × $A_{645}$) (4.86 × $A_{663}$), total chlorophyll ($C_a$ + $C_b$) = (8.02 × $A_{663}$)+ (20.20 × $A_{645}$), carotenoid = [(1,000 × $A_{663}$) (3.27 × $C_a$) (104× $C_b$)]/229.

## Assessing leaf relative water content

Leaves were cut from the plant, weighed immediately (fresh weigh, FW), floated in water for 24 h to achieve turgidity (saturated weight, SW), then oven-dried at 70 °C and weighed again (dry weight, DW). RWC of leaves was calculated according to the formula: [(FW DW)/(SW DW)] ×100%.

## Soluble sugar, sucrose, fructose and glucose analysis

Soluble sugars, sucrose, fructose and glucose were determined using the methods of *Shanghai Institute of Plant Physiology (1999)*. Powdered dry samples (0.05 g) were extracted with four mL of 80% ethanol in a water bath at 80 °C for 30 min. The residue was re-extracted twice, and the three extracts were combined. The supernatants were used to determine the soluble sugars, sucrose, fructose and glucose concentrations.

For determination of soluble sugars, one mL of the filtered solution was mixed with 0.5 mL anthrone solution (1 g anthrone dissolved in 50 mL ethyl acetate) and five mL $H_2SO_4$ and heated at 90 °C for 10 min. The mixture was immediately placed in tap water to cool and the absorbance measured at 620 nm. The concentrations of total soluble sugar were determined using glucose as the standard.

For determination of sucrose, 0.1 mL of the filtered solution was mixed with 50 µL of 2 M NaOH and incubated at 100 °C for 5 min, after which it was immediately cooled. The mixture was added to 0.7 mL of 30% HCl and 0.2 mL of 0.1% resorcinol, heated in a water bath at 80 °C for 10 min and after cooling absorbance was determined at 480 nm. The sucrose concentration was calculated using sucrose as the standard.

For determination of fructose, 0.1 mL of the filtered solution was mixed well with 0.2 mL of 0.1% resorcinol and 0.7 mL $H_2O$, heated in a water bath at 80 °C for 10 min and cooled. Absorbance at 480 nm was determined. The fructose concentration was measured using fructose as the standard.

For analysis of glucose concentration, four mL of enzyme solution including 1 mg o-dianisidine dihydrochloride, 0.1 mg horseradish peroxidase (CAS:9003-99-0, Solarbio) and 1 µL glycose oxidase (1,000 U/mL, dissolved in 0.1 M acetate buffer, pH 5.5, CAS:9001-37-0, Solarbio) was placed at 30 °C until equilibrium after which two mL of the extracted sugar solution was added and mixed well for 5 min at 30 °C. The reaction was stopped by addition of eight mL of 10 M $H_2SO_4$, and the absorbance was determined at 460 nm. The glucose concentration was determined using glucose as the standard.

## Statistical analysis

Statistical analyses were performed with DPS software, Version 9.50. Data were presented as the mean $\pm$ SE and means were compared by Fishers Least Significant Differences (LSD) test at 0.05 probability level. Homoscedasticity was evaluated by Levene's test after some data were logarithmic, sine or cosine transformed. Correlation analysis was conducted between indicators measured and temperature using Spearman's rho method (SPSS 16).

## RESULTS

### Root metabolic activity

Highly significant ($P < 0.001$) effects of *Epichloë* endophyte, calcium nitrate treatment, or treatment time were found on root metabolic activity of *F. sinensis*. In addition, there were some interactions between *Epichloë* endophyte, calcium nitrate treatment, and treatment time for root metabolic activity ($P < 0.001$, Table 1). As shown in Table 2 and Fig. 2, the E+ plants had higher root metabolic activity compared to the E- plants before

Zhou et al. (2021), *PeerJ*, DOI 10.7717/peerj.10568

Peer**J**

**Table 1** Results of split-split-plot ANOVA for the effects of *Epichloë* endophyte (E), calcium nitrate treatment, and treatment time (T) on root metabolic activity, leaf relative water content, chlorophyll and carotenoid contents of *F. sinensis* under cold field conditions.

| Level | Source | df | Root metabolic activity | | Relative water content of leaf | | Total chlorophyll | | Chlorophyll a/b ratio | | Carotenoid | |
|---|---|---|---|---|---|---|---|---|---|---|---|---|
| | | | F | p | F | p | F | p | F | p | F | p |
| Whole plot | Ca(NO$_3$)$_2$ | 2 | 460.247 | 0.000 | 10.425 | 0.026 | 254.385 | 0.000 | 12.557 | 0.019 | 18.098 | 0.010 |
| Split plot | E | 1 | 168.300 | 0.000 | 13.744 | 0.010 | 0.046 | 0.838 | 2.779 | 0.147 | 0.042 | 0.845 |
| | Ca(NO$_3$)$_2$ ×E | 2 | 40.292 | 0.000 | 5.565 | 0.043 | 11.682 | 0.009 | 2.099 | 0.204 | 10.575 | 0.011 |
| | T | 2 | 1047.777 | 0.000 | 373.569 | 0.000 | 857.792 | 0.000 | 141.541 | 0.000 | 32.000 | 0.000 |
| Split-split plot | Ca(NO$_3$)$_2$ ×T | 4 | 60.018 | 0.000 | 5.572 | 0.003 | 120.014 | 0.000 | 1.651 | 0.194 | 29.495 | 0.000 |
| | E ×T | 2 | 16.922 | 0.000 | 58.827 | 0.000 | 31.830 | 0.000 | 4.687 | 0.019 | 15.141 | 0.000 |
| | Ca(NO$_3$)$_2$ ×E ×T | 4 | 14.402 | 0.000 | 3.946 | 0.013 | 37.855 | 0.000 | 1.574 | 0.214 | 9.478 | 0.000 |

Zhou et al. (2021), *PeerJ*, DOI 10.7717/peerj.10568

**Table 2  Results of one-way ANOVA for the effects of treatment time or calcium nitrate treatment on root metabolic activity, leaf relative water content, chlorophyll and carotenoid contents of *F. sinensis* under cold field conditions.**

| Treatment | df | Root metabolic activity | | Relative water content of leaf | | Total chlorophyll | | Chlorophyll a/ b ratio | | Carotenoid | |
|---|---|---|---|---|---|---|---|---|---|---|---|
| | | F | p | F | p | F | p | F | p | F | p |
| | 0 d | 5 | 30.759 | 0.000 | 0.861 | 0.534 | 32.396 | 0.000 | 1.239 | 0.350 | 1.198 | 0.367 |
| Time | 14 d | 5 | 22.468 | 0.000 | 8.025 | 0.002 | 10.784 | 0.000 | 1.114 | 0.403 | 4.535 | 0.015 |
| | 28 d | 5 | 169.141 | 0.000 | 15.491 | 0.000 | 28.544 | 0.000 | 1.126 | 0.398 | 8.233 | 0.001 |
| | 0 mM | 5 | 64.093 | 0.000 | 47.761 | 0.000 | 20.323 | 0.000 | 6.549 | 0.004 | 16.858 | 0.000 |
| Ca(NO$_3$)$_2$ | 25 mM | 5 | 108.670 | 0.000 | 1.317 | 0.321 | 116.833 | 0.000 | 2.802 | 0.067 | 13.073 | 0.000 |
| | 50 mM | 5 | 250.528 | 0.000 | 52.350 | 0.000 | 17.645 | 0.000 | 10.407 | 0.000 | 4.991 | 0.011 |

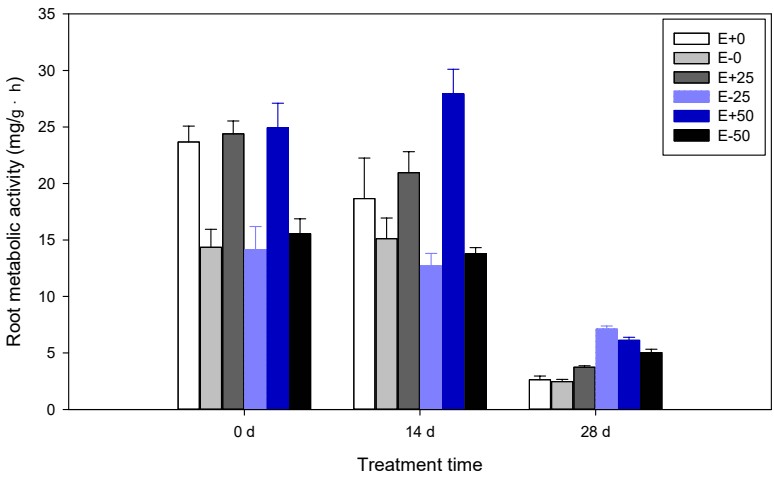

**Figure 2** **Root metabolic activity of *F. sinensis* with and without *Epichloë* endophyte under different Ca(NO$_3$)$_2$ treatments during the study period.** Data represent means $\pm$ standard error (SE). Means within each graph followed by different lower case letters differ statistically among a given treatment time ($P < 0.05$). Means within each graph followed by different upper case letters differ statistically under the same calcium nitrate treatment ($P < 0.05$). E+0, E+25, and E+50 represent endophyte-infected (E+) plants irrigated with Ca(NO$_3$)$_2$ solution at the three concentrations 0, 25 or 50 mM, respectively. E-0, E-25, and E-50 represent non-infected (E–) plants irrigated with Ca(NO$_3$)$_2$ solution at the three concentrations 0, 25 or 50 mM, respectively.

treatment with calcium nitrate ($P < 0.05$). On the 14th day, the presence of endophyte increased root metabolic activity comparing the E+ vs. E- plants, and E+ plants with 50 mM Ca(NO$_3$)$_2$ treatment [50Ca(NO$_3$)$_2$] had the highest level of root metabolic activity among all treatments. On the 28th day root metabolic activities of E- with 25 mM Ca(NO$_3$)$_2$ treatment [25Ca(NO$_3$)$_2$] were significantly higher than others ($P < 0.05$), and significant differences in E+ or E- plants were also observed ($P < 0.05$). When analyzing time shifts in root metabolic activities, samples except for E+ plants with 50Ca(NO$_3$)$_2$ had significantly higher root metabolic activity on the 14th day than on the 28th day, and lower on the 14th day than before treatment ($P < 0.05$).

## Chlorophyll and carotenoid contents

Highly significant ($P < 0.001$) effects of calcium nitrate treatment and treatment time were detected on total chlorophyll of *F. sinensis*. In addition, there were some interactions between *Epichloë* endophyte or calcium nitrate treatment, and treatment time for total chlorophyll ($P < 0.001$, Table 1). There were significant ($P < 0.05$) effects of *Epichloë* status on chlorophyll contents before treatment with calcium nitrate (Table 3). Similarly, on the 14th day, the chlorophyll contents of E+ plants with 25Ca(NO$_3$)$_2$ were significantly ($P < 0.05$) higher than either E- plants or E+ plants with 50Ca(NO$_3$)$_2$. Furthermore, calcium nitrate addition significantly enhanced the chlorophyll contents in E- plants. On the 28th day the chlorophyll contents of E- plants with 25Ca(NO$_3$)$_2$ were the highest among all plants, and the chlorophyll contents of E+ plants supplied with 50Ca(NO$_3$)$_2$ were significantly higher than the other E+ plants ($P < 0.05$). In addition, there was time effect
**Table 3 Chlorophyll and carotenoid contents of *F. sinensis* with and without *Epichloë* endophyte under two different Ca(NO₃)₂ treatments during the study period.**

| Day (d) | Calcium nitrate treatment (mM) | Plant | Total chlorophyll (mg/g) | Chlorophyll a/b ratio | Carotenoid (mg/g) |
|---|---|---|---|---|---|
| | 0 | E+ | 1.85 ± 0.05[aA] | 3.47 ± 0.08[aBC] | 0.0758 ± 0.0094[aB] |
| | | E− | 1.55 ± 0.06[bB] | 3.51 ± 0.05[aBC] | 0.0873 ± 0.0091[aA] |
| 0 | 25 | E+ | 1.88 ± 0.07[aB] | 3.46 ± 0.11[aA] | 0.0836 ± 0.0077[aB] |
| | | E− | 1.56 ± 0.01[bB] | 3.50 ± 0.17[aA] | 0.0779 ± 0.0157[aB] |
| | 50 | E+ | 1.93 ± 0.05[aA] | 3.47 ± 0.12[aA] | 0.0916 ± 0.0088[aA] |
| | | E− | 1.62 ± 0.05[bB] | 3.64 ± 0.03[aA] | 0.0762 ± 0.0089[aB] |
| | 0 | E+ | 1.45 ± 0.15[cBC] | 3.64 ± 0.19[aAB] | 0.0833 ± 0.0113[aB] |
| | | E− | 1.47 ± 0.08[cBC] | 3.86 ± 0.16[aA] | 0.1117 ± 0.0161[abA] |
| 14 | 25 | E+ | 2.08 ± 0.10[aA] | 3.77 ± 0.06[aA] | 0.1408 ± 0.0087[aA] |
| | | E− | 1.81 ± 0.06[abA] | 3.76 ± 0.21[aA] | 0.1246 ± 0.0174[abA] |
| | 50 | E+ | 1.69 ± 0.15[bcA] | 3.64 ± 0.15[aA] | 0.1003 ± 0.0282[bcA] |
| | | E− | 1.76 ± 0.15[bA] | 3.82 ± 0.07[aA] | 0.1258 ± 0.0103[abA] |
| | 0 | E+ | 0.86 ± 0.12[cCD] | 2.95 ± 0.42[aC] | 0.1460 ± 0.0092[aA] |
| | | E− | 0.76 ± 0.03[cD] | 2.78 ± 0.33[aC] | 0.0746 ± 0.0132[bB] |
| 28 | 25 | E+ | 0.74 ± 0.07[cC] | 2.73 ± 0.44[aA] | 0.0659 ± 0.0111[bC] |
| | | E− | 1.64 ± 0.10[aB] | 3.15 ± 0.16[aA] | 0.1374 ± 0.0262[aA] |
| | 50 | E+ | 1.10 ± 0.16[bB] | 3.08 ± 0.23[aB] | 0.0517 ± 0.0184[bB] |
| | | E− | 1.27 ± 0.16[bB] | 3.09 ± 0.20[aB] | 0.1487 ± 0.0541[aA] |

**Notes.**

E+ and E- represent endophyte-infected and non-infected plants, respectively. Data represent means ± standard error (SE). Means followed by different lowercase letters differ statistically among a given treatment time ($P < 0.05$). Means followed by different uppercase letters differ statistically under the same calcium nitrate treatment ($P < 0.05$).

on chlorophyll contents (Table 2). Chlorophyll contents in both control and treated-plants showed significant changes throughout the experiment (Table 3, $P < 0.05$). The chlorophyll contents for the control and 50Ca(NO₃)₂-treated E- plants, and 50Ca(NO₃)₂-treated E+ plants were much lower on the 28th day than before and after 14 days treatment ($P < 0.05$), however the chlorophyll contents in 25Ca(NO₃)₂-treated E- plants were greater on the 14th day than on the 28th day and before treatment ($P < 0.05$).

Treatment time was the only highly significant ($P < 0.001$) effect detected on chlorophyll a/b ratio for *F. sinensis* (Table 1). Apart from that, not all of the chlorophyll a/b ratios in plants on the 28th day were much lower than on the 14th day and before treatment (Tables 2 and 3, $P < 0.05$).

Highly significant effect of treatment time was detected for carotenoids for *F. sinensis*. In addition, there were some interactions between *Epichloë* endophyte, calcium nitrate treatment, and treatment time for carotenoid content ($P < 0.001$, Table 1). On the 14th day carotenoid contents of E+ plants with 25Ca(NO₃)₂ were significantly greater than those of untreated E+ plants ($P < 0.05$). Similarly on the 28th day, carotenoid contents of the control E+ plants were much greater than those of treated E+ plants and control E- plants, moreover calcium nitrate addition significantly increased carotenoid contents of the E- plants ($P < 0.05$). A significant time effect on carotenoid content was also observed ($P < 0.05$). Carotenoid contents of treated E- plants before treatment were much lower

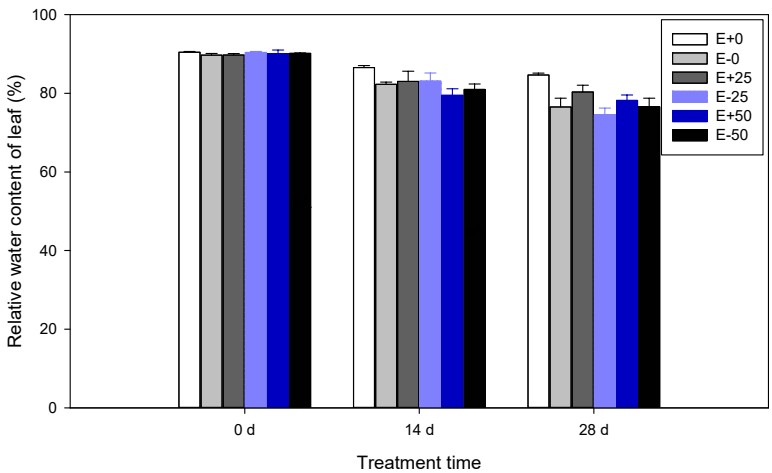

**Figure 3** **Relative water content of leaf in *F. sinensis* with and without *Epichloë* endophyte under different Ca(NO₃)₂ treatments during the study period.** Data represent means ± standard error (SE). Means within each graph followed by different lower case letters differ statistically among a given treatment time ($P < 0.05$). Means within each graph followed by different upper case letters differ statistically under the same calcium nitrate treatment ($P < 0.05$). E+0, E+25, and E+50 represent endophyte-infected (E+) plants irrigated with Ca(NO₃)₂ solution at the three concentrations 0, 25 or 50 mM, respectively. E-0, E-25, and E-50 represent non-infected (E–) plants irrigated with Ca(NO₃)₂ solution at the three concentrations 0, 25 or 50 mM, respectively.

than on the 14th day and 28th day ($P < 0.05$), control E- and E+ with 50Ca(NO₃)₂ for carotenoid contents were significantly ($P < 0.05$) decreased on the 28th day compared to on the day zero and the 14th day, and the carotenoid contents in control E+ sharply increased on the 28th day.

## Leaf relative water content (RWC)

A highly significant ($P < 0.001$) effect of treatment time was detected for relative water content of *F. sinensis*. In addition, there was significant *Epichloë* endophyte-by-treatment time interactions for RWC ($P < 0.001$, Table 1). Relative water contents of leaves were observed to decline significantly ($P < 0.05$) as time passed after treatment (Table 2, Fig. 3). There were marked water increases in control E+ compared with control E- leaves on the 14th day and 28th day ($P < 0.05$). On the 14th day RWC of E+ plants with 50Ca(NO₃)₂ were significantly ($P < 0.05$) less than those of the control E+ plants. In addition, on the 28th day RWC of E+ plants with 0 and 25Ca(NO₃)₂ were maintained at a significantly ($P < 0.05$) higher level than those of other plants.

## Soluble carbohydrates

Highly significant ($P < 0.001$) effects of treatment time or *Epichloë* endophyte were detected for soluble sugars in the shoots or roots of *F. sinensis*. In addition, there were some interactions between *Epichloë* endophyte, calcium nitrate treatment, and treatment time for soluble sugars in the shoots ($P < 0.001$, Table 4). Before calcium nitrate treatment soluble sugars in the shoots were not significantly different across all treatments, whereas soluble sugars in the roots were greater in E+ plants compared with E- plants ($P < 0.05$,

Zhou et al. (2021), *PeerJ*, DOI 10.7717/peerj.10568

**Table 4  Results of split-split-plot ANOVA for the effects of *Epichloë* endophyte (E), calcium nitrate treatment, and treatment time (T) on soluble carbohydrates of shoot and root in *F. sinensis* under cold field conditions.**

| Level | Source | df | Soluble sugar concentration | | | | Sucrose concentration | | | | Fructose concentration | | | | Glucose concentration | | | |
|---|---|---|---|---|---|---|---|---|---|---|---|---|---|---|---|---|---|---|
| | | | Shoot | | Root | | Shoot | | Root | | Shoot | | Root | | Shoot | | Root | |
| | | | F | *p* | F | *p* | F | *p* | F | *p* | F | *p* | F | *p* | F | *p* | F | *p* |
| Whole plot | $Ca(NO_3)_2$ | 2 | 2.418 | 0.205 | 1.801 | 0.277 | 0.656 | 0.567 | 3.150 | 0.151 | 2.621 | 0.187 | 0.632 | 0.577 | 2.096 | 0.238 | 2.435 | 0.203 |
| Split plot | E | 1 | 87.410 | 0.000 | 2.740 | 0.149 | 0.062 | 0.812 | 0.904 | 0.378 | 0.724 | 0.428 | 7.624 | 0.033 | 1.369 | 0.286 | 0.008 | 0.934 |
| | $Ca(NO_3)_2 \times E$ | 2 | 10.056 | 0.012 | 2.311 | 0.180 | 11.669 | 0.009 | 8.116 | 0.020 | 1.389 | 0.319 | 9.016 | 0.016 | 4.902 | 0.055 | 0.102 | 0.905 |
| | T | 2 | 563.155 | 0.000 | 995.675 | 0.000 | 33.611 | 0.000 | 1716.937 | 0.000 | 68.166 | 0.000 | 47.395 | 0.000 | 59.602 | 0.000 | 104.878 | 0.000 |
| Split-split plot | $Ca(NO_3)_2 \times T$ | 4 | 4.875 | 0.005 | 1.553 | 0.219 | 0.356 | 0.837 | 3.489 | 0.022 | 2.094 | 0.113 | 3.550 | 0.021 | 1.659 | 0.192 | 2.631 | 0.059 |
| | $E \times T$ | 2 | 47.415 | 0.000 | 9.319 | 0.001 | 0.494 | 0.616 | 2.482 | 0.105 | 0.484 | 0.622 | 10.790 | 0.0005 | 1.745 | 0.195 | 0.006 | 0.995 |
| | $Ca(NO_3)_2 \times E \times T$ | 4 | 16.478 | 0.000 | 1.151 | 0.357 | 3.654 | 0.018 | 5.023 | 0.004 | 1.821 | 0.158 | 4.481 | 0.008 | 4.582 | 0.007 | 0.097 | 0.982 |

Tables 5–7). On the 14th day E+ plants with 25Ca(NO$_3$)$_2$ significantly accumulated soluble sugars in the shoots ($P < 0.05$) compared to E- plants. On the 28th day the E+ plants had significantly higher soluble sugar concentrations in the shoots than the E- plants under control and 25Ca(NO$_3$)$_2$ ($P < 0.05$). However, calcium nitrate application greatly elevated soluble sugars in the roots in E- plants. In addition, there were substantial time changes in soluble sugar concentrations (Tables 6 and 7). The soluble sugar concentrations of E- plants, control and 50Ca(NO$_3$)$_2$ treated E+ plants were much higher on the 28th day than before and after 14 days treatment ($P < 0.05$). Soluble sugar concentrations for 25Ca(NO$_3$)$_2$ treated E+ plants peaked on the 14th day ($P < 0.05$).

The highly significant ($P < 0.001$) effects of treatment time were detected for sucrose levels of shoots and roots of *F. sinensis*. In addition, there were some significant interactions between *Epichloë* endophyte, calcium nitrate treatment, or treatment time for sucrose levels of shoots or roots ($P < 0.05$, Table 4). The sucrose levels of shoots and roots presented in Tables 6 and 7. After 14 d of calcium nitrate treatment sucrose concentrations of E+ shoots were significantly enhanced by 25Ca(NO$_3$)$_2$ treatment when compared with control E- shoots ($P < 0.05$). On the 28th day sucrose concentrations of E+ shoots with 25Ca(NO$_3$)$_2$ were significantly greater than those of the control or 25Ca(NO$_3$)$_2$ treated E- plants ($P < 0.05$). In addition, on the 28th day control E+ roots accumulated higher sucrose concentrations compared to those of other plants, and E+ with 25Ca(NO$_3$)$_2$ increased more rapidly compared to those of E+ with 50Ca(NO$_3$)$_2$ ($P < 0.05$). With respect to the time series, the sucrose concentrations of roots were higher on the 28th day than on the 14th day and before treatment ($P < 0.05$).

The highly significant ($P < 0.001$) effects of treatment time were detected for fructose concentrations of shoots and roots of *F. sinensis*. In addition, there were some interactions between *Epichloë* endophyte and treatment time for fructose concentrations of roots ($P < 0.001$, Table 4). On the 14th day exogenous calcium nitrate significantly increased fructose concentrations of roots in E- plants ($P < 0.05$, Tables 5–7). Fructose concentrations of plants shoots were significantly different across the time points (before treatment, on the 14th and 28th day). Fructose concentrations of E+ shoots treated were significantly higher on the 14th day than before treatment and on the 28th day, control E- before treatment were lower than on the 14th day and 28th day ($P < 0.05$). The fructose concentrations of roots was significantly less on the 14th day than before treatment or on the 28th day ($P < 0.05$).

The highly significant ($P < 0.001$) effects of treatment time were detected on the glucose concentrations of shoots and roots of *F. sinensis* (Table 4). On the 28th day glucose concentrations of control E+ shoots were significantly higher than those of control E- and E+ with 25Ca(NO$_3$)$_2$ ($P < 0.05$, Table 6). Regarding given treatment time point variance, glucose concentrations of both E- roots supplied with Ca(NO$_3$)$_2$ and control E+ were higher on the 28th day than on the 14th day and before treatment ($P < 0.05$), while other roots in glucose concentrations on the 14th day were much more than before treatment, and lower than on the 28th day ($P < 0.05$).

Zhou et al. (2021), *PeerJ*, DOI 10.7717/peerj.10568

**Table 5** **Results of one-way ANOVA for the effects of treatment time or calcium nitrate treatment on soluble carbohydrates of shoot and root in *F. sinensis* under cold field conditions.**

| Treatment | | df | Soluble sugar concentration | | | | Sucrose concentration | | | | Fructose concentration | | | | Glucose concentration | | | |
|---|---|---|---|---|---|---|---|---|---|---|---|---|---|---|---|---|---|---|
| | | | Shoot | | Root | | Shoot | | Root | | Shoot | | Root | | Shoot | | Root | |
| | | | F | p | F | p | F | p | F | p | F | p | F | p | F | p | F | p |
| Time | 0 d | 5 | 0.623 | 0.686 | 2.824 | 0.065 | 0.873 | 0.527 | 1.148 | 0.388 | 0.774 | 0.587 | 0.696 | 0.637 | 0.201 | 0.956 | 1.199 | 0.366 |
| | 14 d | 5 | 35.378 | 0.000 | 1.191 | 0.370 | 1.438 | 0.280 | 0.445 | 0.809 | 0.370 | 0.859 | 16.387 | 0.000 | 2.005 | 0.150 | 0.052 | 0.998 |
| | 28 d | 5 | 15.033 | 0.000 | 4.331 | 0.017 | 2.740 | 0.071 | 6.057 | 0.005 | 0.071 | 0.996 | 1.368 | 0.303 | 3.991 | 0.023 | 1.857 | 0.176 |
| Ca(NO$_3$)$_2$ | 0 mM | 5 | 45.897 | 0.000 | 114.614 | 0.000 | 12.323 | 0.000 | 581.967 | 0.000 | 11.744 | 0.000 | 27.265 | 0.000 | 20.471 | 0.000 | 35.751 | 0.000 |
| | 25 mM | 5 | 107.412 | 0.000 | 94.298 | 0.000 | 7.913 | 0.002 | 106.278 | 0.000 | 16.352 | 0.000 | 7.122 | 0.003 | 13.305 | 0.000 | 11.899 | 0.000 |
| | 50 mM | 5 | 173.728 | 0.000 | 181.708 | 0.000 | 1.349 | 0.309 | 558.889 | 0.000 | 7.392 | 0.002 | 5.099 | 0.010 | 5.852 | 0.006 | 7.043 | 0.003 |
**Table 6  Soluble carbohydrates in shoot of *F. sinensis* with and without *Epichloë* endophyte under different Ca(NO₃)₂ treatments during the study period.**

| Day (d) | Calcium nitrate (mM) | Plant | Soluble sugar concentration (mg/g) | Sucrose concentration (mg/g) | Fructose concentration (mg/g) | Glucose concentration (mg/g) |
|---|---|---|---|---|---|---|
| 0 | 0 | E+ | 13.90 ± 0.41[aB] | 2.14 ± 0.05[aB] | 9.46 ± 0.84[aC] | 1.70 ± 0.20[aC] |
| | | E− | 14.22 ± 0.27[aB] | 2.14 ± 0.08[aA] | 8.57 ± 0.33[aC] | 1.73 ± 0.08[aA] |
| | 25 | E+ | 13.90 ± 0.47[aC] | 2.16 ± 0.08[aB] | 8.90 ± 0.67[aB] | 1.66 ± 0.13[aA] |
| | | E− | 14.30 ± 0.37[aB] | 2.14 ± 0.03[aA] | 8.68 ± 0.69[aC] | 1.73 ± 0.10[aC] |
| | 50 | E+ | 13.85 ± 0.40[aB] | 2.17 ± 0.06[aA] | 9.46 ± 0.19[aC] | 1.66 ± 0.25[aA] |
| | | E− | 14.04 ± 0.52[aB] | 2.23 ± 0.08[aA] | 9.01 ± 1.07[aC] | 1.76 ± 0.07[aC] |
| 14 | 0 | E+ | 13.39 ± 0.48[bB] | 2.20 ± 0.07[abB] | 10.73 ± 0.24[aAB] | 1.33 ± 0.10[aB] |
| | | E− | 13.00 ± 0.94[bB] | 2.13 ± 0.12[bA] | 12.34 ± 1.54[aA] | 1.38 ± 0.10[aB] |
| | 25 | E+ | 15.52 ± 0.52[aB] | 2.32 ± 0.08[aB] | 14.01 ± 0.84[aA] | 1.58 ± 0.06[aA] |
| | | E− | 13.20 ± 0.10[bB] | 2.17 ± 0.14[abA] | 12.79 ± 0.96[aA] | 1.30 ± 0.18[aB] |
| | 50 | E+ | 13.92 ± 0.18[bB] | 2.28 ± 0.02[abA] | 13.90 ± 2.08[aA] | 1.49 ± 0.23[aA] |
| | | E− | 13.37 ± 0.29[bB] | 2.18 ± 0.14[abA] | 12.23 ± 0.47[aAB] | 1.31 ± 0.08[aB] |
| 28 | 0 | E+ | 18.09 ± 0.28[aA] | 2.69 ± 0.16[abA] | 10.34 ± 0.51[aB] | 2.34 ± 0.06[aA] |
| | | E− | 16.14 ± 0.25[cA] | 2.33 ± 0.13[bcA] | 10.57 ± 0.58[aB] | 1.87 ± 0.23[bA] |
| | 25 | E+ | 18.53 ± 0.52[aA] | 2.69 ± 0.20[aA] | 10.34 ± 1.17[aB] | 1.72 ± 0.03[bA] |
| | | E− | 17.18 ± 0.58[bA] | 2.25 ± 0.15[cA] | 10.68 ± 1.02[aB] | 2.00 ± 0.09[abA] |
| | 50 | E+ | 18.16 ± 0.19[aA] | 2.48 ± 0.32[abcA] | 10.68 ± 1.07[aBC] | 1.96 ± 0.36[abA] |
| | | E− | 18.04 ± 0.39[aA] | 2.64 ± 0.17[abA] | 10.46 ± 1.35[aBC] | 2.12 ± 0.11[abA] |

**Notes.**
E+ and E- represent endophyte-infected and non-infected plants, respectively. Data represent means ± standard error (SE). Means followed by different lowercase letters differ statistically among a given treatment time ($P < 0.05$). Means followed by different uppercase letters differ statistically under the same calcium nitrate treatment ($P < 0.05$).

### Correlation of temperature and indicators measured of *F. sinensis*

Under field conditions, temperature and carotenoid concentration, shoot and root fructose concentrations, or glucose concentration of shoot were poorly correlated (Table 8). There were significantly positive correlation between total chlorophyll or chlorophyll a/b ratio and temperature ($P < 0.05$). The correlativity between temperature and root activity, and relative water content of leaf were significantly higher at the 0.01 level, the correlativity coefficients were 0.734 and 0.761, respectively. However soluble sugar concentration of shoot and sucrose concentration of root negatively correlated with temperature ($P < 0.05$), and soluble sugar concentration of root, sucrose concentration of shoot and glucose concentration of root negatively correlated with temperature ($P < 0.01$) with the Spearman correlations being 0.944, 0.682, and 0.866, respectively.

## DISCUSSION

Low temperature induces phenotypic, physiological and biochemical changes in plants. Low temperature or growth inactivity of the plant may reduce uptake of nitrogen (NH₄NO₃). In this study we compared the changes in physiological performance and soluble sugar concentrations in 5-month-old *F. sinensis* E+ and E- plants when supplied with exogenous Ca(NO₃)₂ to elucidate the mechanism of plant survival during cold winter conditions.

**Table 7 Soluble carbohydrates in root of *F. sinensis* with and without *Epichloë* endophyte under different Ca(NO₃)₂ treatments during the study period.**

| Day (d) | Calcium nitrate (mM) | Plant | Soluble sugar concentration (mg/g) | Sucrose concentration (mg/g) | Fructose concentration (mg/g) | Glucose concentration (mg/g) |
|---|---|---|---|---|---|---|
| 0 | 0 | E+ | 11.29 ± 0.21[aC] | 1.73 ± 0.01[aB] | 6.23 ± 0.47[aAB] | 0.85 ± 0.03[aC] |
| | | E− | 10.40 ± 0.27[bC] | 1.75 ± 0.06[aB] | 5.57 ± 0.67[aABC] | 0.81 ± 0.01[aC] |
| | 25 | E+ | 11.24 ± 0.32[aC] | 1.74 ± 0.08[aB] | 5.57 ± 0.67[aB] | 0.82 ± 0.04[aC] |
| | | E− | 10.42 ± 0.85[abC] | 1.78 ± 0.06[aB] | 5.79 ± 0.69[aAB] | 0.84 ± 0.02[aB] |
| | 50 | E+ | 11.25 ± 0.27[aC] | 1.70 ± 0.03[aB] | 5.68 ± 0.51[aA] | 0.81 ± 0.05[aC] |
| | | E− | 10.37 ± 0.64[bC] | 1.78 ± 0.03[aB] | 6.23 ± 0.88[aA] | 0.86 ± 0.02[aB] |
| 14 | 0 | E+ | 14.65 ± 0.18[aB] | 1.73 ± 0.07[aB] | 4.68 ± 0.19[abC] | 1.05 ± 0.13[aB] |
| | | E− | 14.65 ± 0.80[aB] | 1.75 ± 0.06[aB] | 2.34 ± 0.51[cD] | 1.05 ± 0.07[aB] |
| | 25 | E+ | 14.85 ± 0.11[aB] | 1.76 ± 0.11[aB] | 3.90 ± 0.58[bC] | 1.04 ± 0.15[aB] |
| | | E− | 14.63 ± 0.09[aB] | 1.68 ± 0.07[aB] | 5.34 ± 0.19[aB] | 1.00 ± 0.17[aB] |
| | 50 | E+ | 14.66 ± 0.22[aB] | 1.72 ± 0.06[aB] | 4.34 ± 0.38[abB] | 1.04 ± 0.14[aB] |
| | | E− | 15.20 ± 0.31[aB] | 1.72 ± 0.06[aB] | 4.57 ± 0.58[abB] | 1.05 ± 0.15[aB] |
| 28 | 0 | E+ | 16.67 ± 0.42[bcA] | 3.58 ± 0.09[aA] | 5.40 ± 0.24[abBC] | 1.54 ± 0.23[aA] |
| | | E− | 16.21 ± 0.27[cA] | 3.21 ± 0.05[bcA] | 7.01 ± 1.26[aA] | 1.52 ± 0.05[aA] |
| | 25 | E+ | 16.75 ± 0.31[bcA] | 3.26 ± 0.07[bA] | 5.12 ± 0.19[aB] | 1.32 ± 0.07[aA] |
| | | E− | 17.07 ± 0.59[abA] | 3.27 ± 0.16[bA] | 7.01 ± 1.26[aA] | 1.33 ± 0.14[aA] |
| | 50 | E+ | 16.86 ± 0.50[abcA] | 3.05 ± 0.08[cA] | 5.07 ± 0.24[aAB] | 1.28 ± 0.10[aA] |
| | | E− | 17.61 ± 0.30[aA] | 3.27 ± 0.06[bA] | 6.12 ± 0.84[aA] | 1.28 ± 0.22[aA] |

**Notes.**
E+ and E- represent endophyte-infected and non-infected plants, respectively. Data represent means ± standard error (SE). Means followed by different lowercase letters differ statistically among a given treatment time ($P < 0.05$). Means followed by different uppercase letters differ statistically under the same calcium nitrate treatment ($P < 0.05$).

**Table 8 Spearman's rho correlation coefficients between indicators measured and temperature.**

| Parameter | Temperature |
|---|---|
| Root metabolic activity | 0.734[**] |
| Relative water content of leaf | 0.761[**] |
| Total chlorophyll | 0.577[*] |
| Chlorophyll a/ b ratio | 0.492[*] |
| Carotenoid | −0.203 |
| Soluble sugar concentration of shoot | −0.564[*] |
| Soluble sugar concentration of root | −0.944[**] |
| Sucrose concentration of shoot | −0.682[**] |
| Sucrose concentration of root | −0.564[*] |
| Fructose concentration of shoot | −0.302 |
| Fructose concentration of root | −0.013 |
| Glucose concentration of Shoot | −0.394 |
| Glucose concentration of root | −0.866[**] |

**Notes.**
[*]Correlation is significant at the 0.05 level.
[**]Correlation is significant at the 0.01 level.

Low temperature is known to reduce root vigor (*Sun et al., 2017*). Consistent with this, our results indicated that root metabolic activities of *F. sinensis* that were inhibited by low temperature decreased during prolonged cold stress (Fig. 2, Table 8). Endophyte infection was beneficial for root metabolic activity which was in agreement with results reported by *Liu et al. (2014)* who showed that cucumber inoculated with arbuscular mycorrhizal fungi increase root metabolic activity under low temperature. Further to this, the increase of root metabolic activity was very similar (on 0 d and 14 d) to several previous studies on cucumber and tomato treated with exogenous calcium (*Li, Gao & Liu, 2011*; *Yan et al., 2006*; *Liu et al., 2012*). Furthermore, endophytic fungi or exogenous calcium nitrate may be in effective in promoting root metabolic activity occurred early after cold treatment in field conditions (Fig. 2). In addition, our results showed that E- plants under $25Ca(NO_3)_2$ had higher root metabolic activity than E+25 on 28 d. It is thus likely that endophytes that inhabit aerial grass tissues of *F. sinensis* ineffectively enhanced the root metabolic activity under different $Ca(NO_3)_2$ treatments and low temperature. Ca concentration of roots in the present study need to be studied further.

Exogenous calcium supplied *in vitro* or arbuscular mycorrhiza or PRPP (*Pseudomonas* spp., *Burkholderia phytofirmans*) are capable of alleviating chlorophyll degradation in plants at low temperature (*Schaberg et al., 2011*; *Feng et al., 2010*; *Mishra et al., 2011*; *Zhou et al., 2012*; *Fernandez et al., 2012*; *Zhu, Song & Xu, 2009*). For example, arbuscular mycorrhiza enhanced pigment contents such as chlorophyll a, chlorophyll b, total chlorophyll, and carotenoids in tomato plants subjected to normal or low temperature stress (*Latef & He, 2011*). However it has also been suggested that low carotenoid content was beneficial to pepper species exposed to low temperature and low irradiance (*Ou et al., 2015*). Similarly, *Song et al. (2015)* reported that waterlogging stress induced greater content of chlorophyll in E+ than in E- *Hordeum brevisubulatum*. *Zhou et al. (2012)* concluded that both AMF colonization and paclobutrazol were capable of alleviating the low-temperature damage by increasing chlorophyll synthesis. We found endophyte stimulated chlorophyll levels before treatment with calcium nitrate with inconsistent effects on the carotenoid levels through the time course (Table 3). This may be due to temperature variance affecting fluctuation of endophyte frequency in tall fescue tissue (*Ju et al., 2006*).

In the experiments described in this paper, RWC was significantly reduced with decreasing temperature (Table 8). However, endophyte infection increased the relative water content of leaves on the 14th day and 28th day (Fig. 3). It has been reported that RWC in mycorrhizal and non-mycorrhizal maize leaves are similar at all temperature treatments (*Zhu, Song & Xu, 2009*). One study has shown that inoculation with *Pseudomonas* strains significantly enhances root/shoot biomass, relative water content, chlorophyll, anthocyanin, proline, proteins, and amino acids, which are adaptations to cold stress in wheat (*Mishra et al., 2011*). These findings indicate that RWC may have been increased by microorganisms. Our results showed that the impact of 50 mM $Ca(NO_3)_2$ in reducing RWC was greater for E+ plant than for E- . *Epichloë* endophyte may be playing a role in nutrient uptake and osmotic adjustment.

Soluble carbohydrates appear essential in the response of plants to cold stress with elevated soluble sugar content (*Smith, 1975*; *Trom, Sheath & Bryant, 1989*; *Klotke et*

*al., 2004*; *Stupnikova et al., 2002*). We found significant correlation between soluble carbohydrates and temperature (Table 8). In species as diverse as perennial grass *Miscanthus*, a significant correlation between soluble sugars and minimum temperature has been reported (*Purdy et al., 2015*). The process of cold stress influences carbohydrate concentrations. *Saghfi & Eivazi (2014)* have found that glucose, rhamnose and mannose concentrations vary over time, with the maximum levels on the 6th day, and fructose contents continuously increase in susceptible chickpea whilst in the resistant cultivar glucose, rhamnose and mannose contents increased continuously and maximum levels of fructose were attained on the 4th day. Similarly, the dynamics of the changes in total soluble sugar, glucose, sucrose, sorbitol and fructose concentrations differ in two peach cultivars during the cold winter period (*Yooyongwech et al., 2009*). In the present study, soluble sugar and glucose concentrations of roots continued to increase as time passed, the maximum levels of fructose in shoots were achieved on the 14th day, but soluble sugars, sucrose, glucose concentrations of shoots and sucrose concentrations of roots were observed with maximum on the 28th day (Tables 6 and 7). The results suggest that cold affects the transformation of hexoses into sucrose and its subsequent splitting by invertase into hexoses again. In addition, many plant shoots accumulated more soluble sugars, fructose and glucose compared to roots, however on the 28th day sucrose concentrations in roots increased more than in shoots (Tables 6 and 7). Carbohydrate concentrations between oat and rye tissues have marked differences (*Livingston, Premakumar & Tallury, 2005*; *Livingston, Premakumar & Tallury, 2006*), which is consistent with our finding that soluble carbohydrate concentrations in shoots and roots varied. The concentration of 40 mg/g soluble sugar in *Miscanthus sacchariflorus* at Aberystwyth in November was larger than those observed only 9 mg/g at Harpenden (*Purdy et al., 2015*). These findings demonstrate that the soluble carbohydrate shifts are affected by many factors, such as sugar metabolism and transport, temperature, nutrient availability, stress time, plant tissue, site and species.

As previous works have noted, cold stress (4 °C) induces more carbohydrate accumulations such as starch, glucose, fructose, sucrose, mannose, raffinose and maltose, in grapevine inoculated with *Burkholderia phytofirmans* than in grapevine non-inoculated (*Fernandez et al., 2012*). Our data collected also suggested that *Epichloë* endophyte infection significantly increased soluble sugars of shoots with $25Ca(NO_3)_2$ and fructose of roots under the control conditions on the 14th day, and on the 28th day significantly increased soluble sugar of shoots with control and $25Ca(NO_3)_2$, glucose of shoots and sucrose of roots under the control conditions (Tables 6 and 7). Under low-temperature stress, *Zhu et al. (2015)* have found arbuscular mycorrhiza raise soluble sugars and reducing sugars of leaves, and sucrose and fructose concentrations of roots in maize, and a similar result is also reported by *Chen et al. (2014)*.

Although these studies have tested that microorganisms have an active role in the accumulation of sugars in the host under cold stress, they did not examine as $Ca(NO_3)_2$ application as in our study. Our results also indicated that calcium nitrate addition contributed to the accumulation of soluble sugars, sucrose, fructose or glucose in shoots and roots of both E+ and E- plants during long term cold stress (Tables 6 and 7). Calcium supplementation increases sugar concentrations such as fructose, sucrose, and glucose in

red spruce during both the fall and winter under field conditions (*Halman et al., 2008*; *Hawley et al., 2006*). It is thus likely that amylase activity has been induced by calcium (*Bush et al., 1989*; *Sadhukhan et al., 1990*; *Nielsen, Fuglsang & Westh, 2003*). As far as we know, this is the first study to report that calcium nitrate in combination with a fungal endophyte strongly increased soluble sugars in shoots (Tables 5 and 6).

In addition, *Soto-Barajas, et al. (2015)* suggested that *Epichloë* endophytes may affect nutrient absorption of *Lolium perenne* in field experiments and E+ had significantly lower Ca content in shoot at the flowering stage than E- plants. It thus is logical that supplementation with calcium would further improve plant performance. To the best of our knowledge, the results of the present study are the first evidence of a relationship between calcium nitrate and endophyte in *F. sinensis* exposed to cold field conditions.

# CONCLUSIONS

In this work *F. sinensis* and associated *Epichloë* endophyte were used to study the effect of endophyte infection on physiological performance and soluble sugar concentrations under calcium nitrate and cold field conditions. *Epichloë* endophyte or $Ca(NO_3)_2$ significantly affected root metabolic activity, total chlorophyll, chlorophyll a/b ratio, and carotenoid. *Epichloë* endophyte and $Ca(NO_3)_2$ had no significant effect on chlorophyll a/b ratio. Some soluble sugar concentrations were affected by endophyte infection or interaction of the *Epichloë* endophyte and $Ca(NO_3)_2$ but not by $Ca(NO_3)_2$ alone. Further studies regarding soluble carbohydrate changes in addition to analysis of related enzyme activities, nutrient absorption and gene expression are necessary to elucidate the detailed mechanism of increased cold stress tolerance.

# ACKNOWLEDGEMENTS

We thank Dr. Thiago Oliveira and an anonymous reviewer for their constructive comments on the earlier version, which have significantly improved the article. We also thank CH Guo, JX Cao, GS Bao, SH Chen, TX Chen and LH Xue for trial assistance, and Dr. XZ He (School of Agriculture and Environment, Massey University, NZ) for assistance in data analyses.

## Funding

This work was supported by the National Basic Research Program of China (2014CB138702), the National Natural Science Foundation of China (31760697), the Second Tibetan Plateau Scientific Expedition and Research (STEP) program (2019QZKK0302), the Program for Changjiang Scholars and Innovative Research Team in University, China (IRT17R50), the Fundamental Research Funds for the Central Universities (LZUJBKY-2020-it11) and the 111 Project (B12002). Support was also provided from USDA-NIFA Multistate Project W4147, and the New Jersey Agricultural

Experiment Station. The funders had no role in study design, data collection and analysis, decision to publish, or preparation of the manuscript.

## Grant Disclosures

The following grant information was disclosed by the authors:

National Basic Research Program of China: 2014CB138702.

National Natural Science Foundation of China: 31760697.

Second Tibetan Plateau Scientific Expedition and Research (STEP): 2019QZKK0302.

Program for Changjiang Scholars and Innovative Research Team in University, China: IRT17R50.

Fundamental Research Funds for the Central Universities: LZUJBKY-2020-it11, 111 Project (B12002).

USDA-NIFA Multistate Project: W4147.

New Jersey Agricultural Experiment Station.

## Competing Interests

Richard D. Johnson is employed by AgResearch Limited of New Zealand. The authors declare there are no competing interests.

## Author Contributions

- Lianyu Zhou conceived and designed the experiments, performed the experiments, analyzed the data, prepared figures and/or tables, authored or reviewed drafts of the paper, and approved the final draft.
- Chunjie Li conceived and designed the experiments, authored or reviewed drafts of the paper, and approved the final draft.
- James F. White and Richard D. Johnson analyzed the data, authored or reviewed drafts of the paper, and approved the final draft.

## Data Availability

Raw data are available in the Supplementary Files.

## Supplemental Information

Supplemental information for this article can be found online at http://dx.doi.org/10.7717/peerj.10568#supplemental-information.

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
