# Peer review of "Synergism between calcium nitrate applications and fungal endophytes to increase sugar concentration in Festuca sinensis under cold stress"

_PeerJ, doi:10.7717/peerj.10568_

## Round 0.1 · original submission · Major Revisions

This work has positive estimates from two reviewers. However there is a question about reproducibility. There are a lot of detailed comments to be fixed. But the remarks are rater technical than critical. As academic editor I encourage resubmit the revised manuscript soon.

·

Basic reporting

The article was written in a clear and technically correct English, and a sufficient number of references is supporting how the works fit into the field of study. However, Table 4 is not too clear and potentially will causes confusing in the readers to interpret it. I strong suggest that they reorganize this table into two different ones. All appropriate raw data have been made available, but, due to inadequate statistical description, the same model used by the authors can't be reproduced easily. Please, include the statistical model (with equations).

Experimental design

i) Lines 100-108: They are inducing the reader to understand that they performed a completely randomized design, which does not apply to most of the experiments conducted on the field. More information detailing the experimental structure should be necessary to support the understanding about their experimental design. If the authors had structured in the field a completely randomized design, what were the reasons to apply this approach?

ii) Line 101: Authors have to be more clear if they considered 36 plants by line or a total of 36 plant considering both lines. Just on the line 107 the reader can indirectly understand that a total of 36 plants was used on. Please, make this clear before.

iii) Lines 100-108: The treatment looks a factorial structure, as they combine levels of seeds (infected or not) with levels of Calcium (0, 25, 50), however there is no information about treatment structure. Please, provide this details in the manuscript.

iv) Lines 108-110: As the author described that 12 plants under each level of Ca for either E+ or E- were harvested independently over three time periods. As those plants were established into same plot, there is a spatial relationship between them, consequently, this cannot be viewed as a random sample. How the authors control confounding between plots and treatment on plants?

v) Lines 108-110: As the authors are using a subsample on the plots (12 plants), how they included the effect of plots (sample) and the effect of plant (subsamples) on the statistical analysis? This description will be fundamental to guarantee that their analysis is correctly developed, and, consequently, make their finds more reliable. Probably the effect of plots and subplots should be added in the analysis of variance.

Validity of the findings

The paper can potentially has a high impact into to improve sugar concentration in Festuca sinensis. However, as the authors did not describe their experimental conductions nor statistical model sufficiently, some doubts arise about the validity of the results obtained. I suggest that the authors re-write the Statistical Analysis including details about the approach used and justifying it. Furthermore, all comments described about the experimental design should be clarified in order to prove the validity of their results.

Additional comments

i) Lines 170-176: There is no statistical model provided over the paper. Please, include it as the model can help to understand how the experimental structure was modelled.

ii) Lines 110-111: As the authors measured the temperature over the experiment during day and night. How they included this information in the statistical model?

iii) Lines 113: Could be interesting the inclusion of the most important response variable, categorized by treatment, in the Figure 1, as the interaction between treatment and temperature has an direct effect on the response.

iv) Lines 180-181: The authors could provide, as supplementary material, a plot of the residuals versus fitted values to check error independence assumptions, Q-Q plot of residuals versus theoretical residuals to check the error normality assumptions. In case of completely randomized design, the Shapiro-Wilk test for normality and Levene test to check homoscedasticity could also be provided in order to guarantee the validity of their finds.

v) Lines 188-189: What was the reasons that E- plants under 25 Ca had higher root activity than E+25?

vi) Line 206, 222: Authors should not assume that there were substantial time series effect as they don't have enough information over the time. I encourage the authors to focus in describe the effects considering the time as a factor rather than as continuous.

vii) Lines 257-259: Authors described that exogenous application of calcium in combination with endophyte actively effects soluble sugar accumulation. I can see an possible significant marginal effect of the time on soluble sugar accumulation, but the effect of exogenous calcium on E+ and E- plants are not enough evident based on means and standard errors, and without model assumptions checked. As the authors have a factorial design, should be appropriate that they included, in the supplementary material, the the effects of each factor estimated at each levels of the other factors, to show that they are yielding conclusions that are valid over their experimental conditions.

viii) Lines 312-313: Authors said that "Interestingly, the synergistic effects of endophytic fungi and exogenous calcium on root activity occurred early after cold treatment in natural field conditions (Fig. 2)". What they means about "Natural Condition"? This results could also shows that there is a possible confounding between natural field conditions and treatment effects. How the authors could explain it?

ix) I disagree with the authors about they infer that calcium treatment enhanced the water content of leaves. I would like to see the parameters estimations of the three-way ANOVA, as the Figure 3 suggests the opposite, that means, there are more water content of leaves without calcium treatment.

Reviewer 2 ·

Basic reporting

Generally good, but see comments in last box below
English language would benefit from additional checking

Experimental design

The research question could be better linked to the cited references.
There is a concern that effects attributed to Ca might be a response to N since Ca(NO3)2 was the key treatment.
A map of the spatial arrangements of the treatments with the raw data would be desirable.
Comments below expand on these comments

Validity of the findings

The possible influence of applied nitrate on the results needs to be addressed.
The citation of other work is sophisticated, but the linkage of the authors' work to the cited work is actually quite weak, as indicated by very generalized conclusions in section 5.

Additional comments

PeerJ MS 45909; Zhou et al. Synergism between calcium applications and fungal endophytes to improve sugar concentrations in Festuca sinensis under cold stress
General comments
1. Overall the MS is well prepared. An interesting plant-endophyte symbiotic system was studied and a pleasing range of physiological parameters measured. It is a pleasant change to have the raw data for inspection, and for the most part I was able to replicate the ANOVA Tables presented in Table 1. (However, I think the F-value of treatment time for root activity should be 1454, not 145.4, according to my ANOVA). The English grammar shows clear signs of checking by a native speaker, though some words and sentences requiring attention have been missed, especially in the results section and Figure and Table captions (or adjusted by the Chinese authors after the checked MS was received). The data on root activity differences between E+ and E- plants is novel and interesting. Despite these good points, I do have some significant concerns about the MS as outlined below.

Specific comments
1. Confounding of Ca and N effects: A major concern I have about this piece of research is that the results are presented as a response to calcium, but the treatment was Ca(NO3)2, so plants have received a double stimulus of Ca and N. Inspection of the Methods section shows that the quantity of elemental N added with the Ca treatments was up to 42 mg per plant. I find no mention of plant total dry weight but if plants were of a smallish size this could potentially be in excess of 2% plant DW which would be a major physiological stimulus. I can not see how we can differentiate the Ca response from the N response under the current experiment design. It would seem that to satisfy the aim of measuring a calcium response, either a calcium salt with a plant-inert negative ion would need to be used, or the differential response calculated between the Ca(NO3)2 application and an equivalent amount of N applied without Ca in a form such as urea or NH4NO3. For publication, this issue must be satisfactorily addressed. This would include mention somewhere in the MS of typical plant root and shoot DW at harvest and a logical explanation of the separation of Ca and N effects in the results interpretation, since they are confounded in the treatment application.
2. Quality of research hypothesis: For this reviewer the research hypothesis seems very general and without detail. In my observation, the traditional approach to research in China is to identify the system one wants to work with (in this case a Festuca-Epichloë symbiosis) and then introduce a perturbation (in this case Ca application) and record the response of the experimental system. The results are then published as response observations in juxtaposition with some nominally relevant results of other experiments and speculative or very general mechanistic interpretation by authors (as seen at lines 343 – 367, e.g, leading to a lack of specificity in the closing sentence at lines 387–389). This is different from the Western approach where there is much more focus on hypothesis development, identification of an experiment design to address the hypothesis, and interpretation of results according to physiological mechanisms at play. This reviewer’s impression is that the experiment might have been designed under the traditional Chinese approach and then presented to seem to fit the Western research philosophy. Specifically with respect to the way the introduction builds to the research hypothesis, there is a very general assertion (with copious cited references) that cold retards plant growth and Ca can be a stimulus, but when we come to the research aim at lines 83 – 84, there is as yet no detail at the physiological level of plant processes that may be affected by cold, with the cold response ameliorated by Ca. For example, at lines 72 – 73 we have “These observations have led to the hypothesis that agricultural strategies to assist plant adaptation to low temperature stress include exogenous chemical applications or symbiotic technology”. For this reviewer some detail from the cited references of the exogenous chemicals used in the cited references and their impact on the plant-endophyte interaction and relevance to the hypothesis at the physiology level would be highly desirable. This detail could be constructed so as to give insight into the reasons for the choice of measurement parameters for the experiment (root activity, chlorophyll levels, relative water content, soluble carbohydrates).
3. Experiment field layout, randomisation and statistical analysis validity: Lines 101–102 mention 2 rows of 36 E+ and 36 E– plants per replicate, so that is 216 plants in the experiment plot, but the data have only 54 rows. Therefore it is not clear how the plants for the various treatments were spatially arranged in the field. This is important because valid ANOVA requires randomisation to satisfy the assumption that each data point is independent of the others. Even if it is not published, it would be good that the authors supply to the reviewers or for inclusion with the raw data in the word file, a map showing the special placement of each of the 54 sampled plants and their treatments (Endophyte, Ca and day). For example if Ca0, Ca25 and Ca50 treatments are grouped along a row and the rows have some stronger and weaker patches, then this can lead to the false identification of a treatment effect or interaction effect in the ANOVA. The MS as presented does not show there is a problem (though separation of E+ and E– into different rows is a possible concern and might be better treated as a split plot design, not a factorial ANOVA as used by the authors); but equally it does not confirm there is not a problem, and that would be solved if a map of the spatial organisation of the plants from each treatment is available. Regarding 216 plants and 54 data observations, maybe 4 plants were planted in May for every one needed at harvest in October/November? Meanwhile layout details at lines 107–108 seem to be different from those given in lines 101-102. It highlights the value of providing a map for clearer understanding. Did harvests at days 0, 14 and 28 use the same plants or different plants each time?
4. Use of could in reporting and discussing results: In Chinese science writing use of ‘ke yi’ (可以) is common when reporting or discussing results but in English we like to report the result from the experiment as a fact, without saying ‘can’ or ‘could’. In English ‘could’ is a subjunctive verb form expressing doubt if something will happen. So at line 385, it might be better to choose another phrase instead of “it could be”. (Also line 317.)
5. Low soluble CHO levels: The CHO levels in Table 4 seem low. Is it because the plants are growing in harsh conditions, or could there be an analysis calibration problem?

Annotated reviews are not available for download in order to protect the identity of reviewers who chose to remain anonymous.

---

## Round 0.2 · Minor Revisions

Thank you for the manuscript update. It has better estimates from the reviewers. But both reviewers still have critical comments demanding another reviewing round. Please check the statistical part as suggested by reviewer #2. Advise additional English language editing.

·

Basic reporting

The paper has a clear aim, and part of comments done on the first review were included in the new version. The proposal manuscript has merit to be published, however, some questions still arising through methodology and results that should be clarified. All questions were described in details in the following sections.

Experimental design

Line 106 - Authors said that "the trials were arranged using a split-plot design with E+ and E– as the main plots, Ca(NO 3 ) 2 levels as the subplots, and sampling time as the sub-sub-plot". I agree that model could be viewed as described by the authors, however It should be interesting they describe how the repeated measurements on subject was captured by their model on the methodology, as I've had requested in the first review. For example, an way to model a design like that is through mixed-effects model. Furthermore, as the repeated measurements on subject are probably correlated, they also could explore the individual variability to describe how strongly subjects in the same group resemble each other, using the intraclass correlation coefficient. This will enhance the discussion of the paper as well.

Validity of the findings

1) I requested to the authors, as supplementary material, a plot of the residuals versus fitted values to check error independence assumptions, Q-Q plot of residuals versus theoretical residuals to check the error normality assumptions, and Levene's test to check homoscedasticity in order to guarantee the validity of their finds, however they provide just Levene's test.

The Levene's test shows that most of response variables were not obtained based on random sampling from a population with equal variances (p-value < 0.05). Thus, authors should include a more complex structure in the variance-covariance matrix to correctly fit the model to their database, probably including different variances by treatment. In this sense, I recommend the implementation of a new variance-covariance structure in order to get correct inference and p-values.

2) What do authors mean by "Univariate analysis of variance was conducted tests of between-subject effects according to the general linear model"? This phrase is not clear at all. Please provide the formulation of this general linear model.

3) Table 4 is showing no two-way nor three-way interaction effect for Soluble sugar concentration (root), Fructose concentration (shoot), and Glucose concentration (root and shoot) (see p-value > 0.05). In this sense, please do a stepwise selection procedure for each model in order to select a more parsimonious model for each response variable.

Reviewer 2 ·

Basic reporting

The first submission was generally good; the authors have tried hard to address issues raised by reviewers. The English grammar, especially in revised sections, needs checking by a native English speaker. There are a number of mistakes (e.g. 'more much' at line 343, many 'awkward' sentences, and frequent mismatching of singular and plural adjectives, nouns and verbs.

Experimental design

Both reviewers raised concerns on the first version. The site map now supplied by the authors confirms the reviewers' concerns were valid. The issues raised by the reviewers remain unadressed by the authors. Currently the authors state they used a split-split-plot design, but the present ANOVA Tables in their results are for a three-way factorial design. I have provided extended comment on this question in an attached file. Regardless of which design is chosen for reporting purposes, the authors at least need to present the ANOVA Tables for the design they state in the materials and methods that they used.
To assist the authors, an example ANOVA table for a split-split-plot design, is compared in the attached file with a three-way factorial ANOVA for the authors' root activity data.

Validity of the findings

A critical point in my first review was No.1, 'confounding of Ca and N treatment effects. Since the treatment was Ca(NO3)2, I do not see how we can exclude the possibility that the observed plant physiology responses are to N and not Ca, or to both elements working together? The authors' answer does not satisfy me. I have added further explanation of my concern in the attached file. Perhaps this is now a matter requiring the Editor's decision to settle it.

Annotated reviews are not available for download in order to protect the identity of reviewers who chose to remain anonymous.

---

## Round 0.3 · Minor Revisions

The manuscript has been updated and now looks much better. But both the reviewers still have critical remarks demanding some revision. The questions are about separate action of Ca and N, statistical estimates (see recommendation from reviewer #1). If you feel that a reviewer has biased opinion, we can ask another reviewer, of course. However I believe that the remarks are adequate. Please reply point-by-point. I encourage the authors to fix the remarks and resubmit the manuscript to PeerJ.

·

Basic reporting

I recommend a carefully review about experimental design, statistical methodology and the inclusion of a paragraph to discuss the limitations of this study.

Experimental design

In a new revision of their map of the spatial arrangements of the treatments I found serious experimental design errors. It appear that they systematically distributed the levels of each factor on plots and subplots rather than using a randomization procedure.

One of fundamental principle of experimentation is the randomization, where the probability of a unit receive any treatment should be the same. Furthermore, randomization reduces confounding among observed and unobserved factors. This can puts all considerations and conclusions of this study as being unreliable. There is no way to distinguish if the effect of levels of Ca nor plants, nor interaction between them were different due to a "true" effect of them or possibly because there is a confounding between them and experimental error. In a split-plot design there are still more concerned about error and factor confounding as there are two source of errors, one from the plot and another from split-plot. Perhaps this is now a matter requiring the Editor's decision to settle it.

Validity of the findings

1) My question about validity of their findings was not properly answered by the authors.

Although they are using split-split-plot design, all models have strong assumptions that should be verified before report any findings. Including diagnostic plots, and statistical tests about model assumptions for Reviewers or as supplementary material are fundamental to certify that significant effects, and hypothesis tested are validity. I would like to see a plot between fitted values vs. residual and a statistical test for homoscedasticity.

Reference: Montfomery, D. Design and Analysis of Experiments, John Wiley & Sons, 8 ed., 2013

2) In the last review I requested a model selection procedure because some interactions were not significant, however, they did not addressed this questions. Additionally, In the new Table they reported p-values higher than 1, how they found it as p-value takes value between 0 and 1?

Additional comments

I did not see any paragraph in their manuscript discussing about a possible confounding of Ca and N treatment effects (as mentioned by Reviewer 2), which could be relevant for further studies.

Reviewer 2 ·

Basic reporting

Covered in previous review

Experimental design

With a map of the experiment layout now provided and the ANOVA now presented as a split-split plot design, this reviewer is now happy. We should also not that in many cases the statistical F values are very large so debate about which model to use is less relevant as any model will declare significance.

Validity of the findings

My number one concern was always that the experiment design does not allow any logical separation of the effects of the element calcium and the element nitrogen when adding calcium nitrate as a treatment. In this second revision, the authors continue to avoid any direct answer on this point. The reference cited for work on a tree species is not at all convincing. The purple-colored track changes comments added then removed again near line 93 ('We ignored .. cell.') are revealing. I note, however, that in the conclusions of the later versions, unlike the first submission, the authors now refer to the effect of 'calcium nitrate', instead of simply 'calcium'. This is much more scientifically correct. If the authors will now do the same in the title and refer to synergy with 'calcium nitrate' this reviewer will be satisfied.

In hindsight it occurs to me that the authors very likely still have residual leaf samples from their work. If so, analysis of leaf tissue for Ca and N concentration would likely be possible within days and would quickly resolve whether or not the nitrate component in calcium nitrate could have played a role in generating the results observed.

I consider all other comments and requests have been addressed well. There remain some minor grammar points. For example I think 'a' (indefinite article) inserted before split-split-plot at line 188 would be better. Maybe cosmetic adjustments like this can be carried out by the typesetting team.

---

## Round 0.4 · Minor Revisions

Thank you for the manuscript update. However, both the reviewers still have some critical remarks demanding revision. Please check the experiment map - it is inconsistent with previous figures. Even the results changed now. Please update all the figures then or change the presentation scheme. Tone down the conclusion, if necessary. I believe the paper is almost ready, just due to several revisions the presentation structure was changed losing initial conclusion. At the next review round we will ask new reviewers to avoid pre-justice.

·

Basic reporting

Thanks to the authors for answering all my questions in this session.

Experimental design

Thanks to the authors for answering all my questions in this session. Their experimental design appears to be systematically randomized rather than give the same probability of a whole plot receive a specific dosage of Calcium nitrate and the same happened in subplot. I reccomend the authors consider apply all the principles of experimentation on their next experiments. When one of them are broken we involuntary can introduce biases that sometimes is really difficulte to evaluate in the data analysis process. The model proposed to analyse this experiment, e.g., is not accounting for the systematic process of randomization.

Example: If you observed the lines within plots, there are a systematic inclusion of E+ and E-. This shows that the authors introduced a new factor in the experimental design due to the randomness process adoped. Clearly, the impact of this new factor in the whole experiment is confunded with the treatment effect.

On the other hand, the inclusion of the experimental map help readers to undertand this limitation. If the authors can guarantee that all the phisics and enviromental factors were the same for each plot, thus the effect of this new factor tends to be small.

Validity of the findings

The authors presented the Levene's test for homoscedasticity, The Table S1 shows that for most of their response variables there is a no acceptation of the null hypothesis. This shows that the variance between treatment were not the same. There are some statistical solutions to account for heterocedasticy in the model. However, I initially suggest a variable transformation (log, square root, etc.). Generally, variable transformation affects variance function in order to obtain more similar variances for treatments, and sometimes also improve normality assumption. The correction of the variance issue in fundamental to get comparison between treatments more reliable. I know that the authors have been doing a lot of work in our suggestion, and they are doing a nice work here. However, I see myself obligate to strong recommend this correction in order to improve their results.

In addition, the log transformation of Root activity was not sufficient to stabilize the variance.

I recommend to the authors read the paper titled as 'An Analysis of Transformations' described by Box and Cox in 1964. It could be helpful in addresses this issues.

Additional comments

No more comments here

Reviewer 2 ·

Basic reporting

The Map in FigS1 is incompatible with Tables 1 and 4 in the MS; see below.

Experimental design

I have to apologise that in my response to the second revision I did not notice that the experiment map was changed. Was it mentioned by the authors anywhere? If it was, I missed it, but when reviewing revision 2, I simply referred to the first map which I had downloaded and saved. Never in my previous experience have the researchers changed the map of their experiment during MS review. For this reviewer, the new experiment map now presented as Figure S1 is incompatible with the MS. To explain, I will call the first map supplied with the first revision Map 1 and the new map now supplied with revision 3, Map 2.
In the original submission it was impossible to know the physical layout of plants in the experiment from the information in method and materials so I asked for a map. Figure S1 (Map 1) was supplied by the authors with Revision 1 and the authors called it ‘a map of the spatial arrangements of the treatments’. It showed 3 replicates, each with an E- and an E+ main plot. E- and E+ as reviewer 1 pointed out were not randomised but I was prepared to overlook that even though I would expect a high impact journal to reject a MS for this kind of problem. As reviewer 1 rightly points out, any soil property gradient across the field where the experiment was located would differentially affect E+ and E- in the same way in all three replicates so experimental error from the soil property gradient and treatment effects will be inseparably confounded in this scenario. It is not reasonable to say as the authors did that soil fertility gradients were believed to be controlled by cultivation; firstly because cultivation mixes soil vertically more than horizontally within quite a small area and not over a distance across a field, and secondly because in the trait for which I did a trial ANOVA (previously suppplied), the replicate effect (which the authors did not extract) had a very large sum of squares.
Map 1 was consistent with the statement by the authors that they had used a split-split plot design with Endophyte as the main plot effect and CaNO3 as the split plot effect. At the main plot or whole plot level there are 6 plots (E+/E- x 3 replicates) (though unrandomised). For CaNO3 there are 3 plots within each mainplot (0, 25, 50) and those are randomly allocated – their spatial position is different in each replicate. In this way there are 6 main/Split plot combinations with 3 replicates = 18 obvious plots in the experiment. Each of those 18 obvious plots has 12 plants. We are not told exactly how the Time effects (0, 14, 28 days) were allocated to plants but that is perhaps a minor point.
It is believable that researchers inexperienced in statistics might first analyse Map 1 as a three-factor factorial design. The non randomisation of E+ and E- suggests it was not conceived as a split plot design but the three factors are obviously and clearly all present so a factorial design could seem logical. But if it was only a ‘diagram’ that did not accurately show the plot positions, why give a diagram when a map was asked for and why label it a ‘map of the spatial arrangement of the treatments’?

If we analyse Map 2 which I have become aware of only while reviewing revision 3, we see that the columns labelled ‘Whole plot’ (I, II, III) are actually replicates and that within these we see the main plot factor is actually CaNo3 (3 randomised main plots per replicate) and the split plot factor is E+/E- which is allocated differently within each CaNO3 plot. Each E+ or E- split plot is a strip of 6 plants that provides for 2 plants for each of three times (0, 14, 28 d) in each of three replicates. However there are 4 strips, not 2 strips in each split plot so the authors have not stated the reason for the extra plants. Was it to have spare plants to ensure healthy plants were always used for measurement, or was some other research done with the other plants for publication elsewhere? If other research it is good to be transparent and tell the readers about it.

If Map 2 is the true map of the experiment then in my opinion all the ANOVA tables have to be recalculated with CaNo3 as the main plot factor and E+/E- as the split plot factor. All the F values and P values will change! (This applies to Table 1 and Table 4 in the MS.)

It is not so easy for this reviewer to believe that if the researchers carefully laid out the experiment in the field as in Map 2 for a split-split-plot design, they would then present their MS with a simple factorial analysis, or that they would make a mistake now by wrongly identifying which was the main plot factor and which was the split plot factor in their Map2. That said, this reviewer is human, so maybe I am making the mistake! I wonder if the United States and New Zealand authors or their statistics colleagues have had any input to the statistical decisions in this data analysis and/or if they have inspected and approved this part of the work?

Validity of the findings

When the map of plot layout shows CaNO3 as the main plot factor and E+/E- as the split plot factor, it is necessary that the ANOVAs in the Tables follow the same.

In the trait I prepared an ANOVA for the replicate effect seems more important than the authors recognised and maybe should be extracted in the main plot stratum of the ANOVA.

Additional comments

Other points:
As reviewer 1 points out, a P value by definition can not be greater than 1. Even so the authors still keep two P values of 1.019 and 1.027 in their Table 4. It would be good to fix this problem.

I remain concerned that the total soluble sugar levels of shoots at 14 – 18 mg/g are lower than I would expect for grass shoots, unless there is a major proportion of dead material in the sample. I looked for another reference on soluble sugars in Festuca pratensis and I found a paper of Wang et al. (also from Lanzhou University) in Functional Plant Biology 2017 where the stated levels are 6 – 12 µg/g (less than the free proline content). At least I am certain those other authors would not be correct, but it does not help provide a benchmark to know if these values are reasonable. Since the authors already gave assurance on this point I will not pursue it any further.

I can’t help noting that in the first submission of this MS the authors were sure the Ca component of the CaNO3 treatment was the cause of the treatment effects. Now they allow that it could be a nitrogen effect and they carefully avoid any discussion about whether there is evidence in favour of one or the other. So it is a different paper now, with endophyte effect on root activities in Fig.2 now the most interesting part.

---

## Round 0.5 · Minor Revisions

We have the detailed comments on the text from second reviewer indicating minor revision. The comments are in PDF attached by the reviewer. Please consider, update and resubmit the manuscript.

Reviewer 2 ·

Basic reporting

As this is version 4 we only need to deal now with the few remaining issues.

Experimental design

As this is version 4 we only need to deal now with the few remaining issues.

Validity of the findings

As this is version 4 we only need to deal now with the few remaining issues.

Additional comments

It is appreciated that the authors have patiently addressed all that was asked of them. This reviewer respects that and does not wish to ask any more.
There remain now a few (maybe 10) cosmetic issues around the description of the design:
1. On the map (Fig. S1) the titles 'whole plot i .. iii on the top row should be replaced with 'Replicate i .. iii' and the statistical analysis section within M&M revised as follows (see also uploaded PDF with comments): The trials were arranged using a split-split-plot design with a total of 216 plants of F. sinensis, where three Ca(NO3)2 concentrations were randomly assigned to three replicates as main plot treatments, two endophyte statuses (i.e., E+ and E–) were randomly assigned into the main plots as spilt plot treatments, and randomly selected plants were sampled at three times in two strips (two plants each time) as the split-split plot s in two strips (Fig. S1). There were 18 treatments (2 endophyte types × 3 Ca(NO3)2 concentrations × 3 sampling times), with three repetitions for each treatment and 12 plants for each repetition (i.e., 36 plants for each treatment). There was a distance of 22 cm between strips (i.e., endophyte types) and 18 cm between plants within a strip. For spilt plots, plot size was 20 m2 with two surrounding-protection strips and two sampling strips. The strips of plants were 0.5 meters apart from each other.
The headings of the various Tables and the identification of the ANOVA effects in column 1 of the Tables, inconsistently use 'Ca' and Ca(NO3)2. Each Table heading and column1 identifying the ANOVA effects should be carefully checked and made consistent. The identifier should not be 'Ca' because N is also involved in the treatment. At least the identifier should include 'Ca' and 'N', and be consistent across all Table headings, table content and manuscript text.
The titles for the supplementary data files should be edited carefully - there are mispellings of words like replicate and spaces where spaces should not be.

Annotated reviews are not available for download in order to protect the identity of reviewers who chose to remain anonymous.

---

## Round 0.6 · accepted · Accept

Thanks for the patience during multiple manuscript updates. All the comments were taken into account. I endorse manuscript publication in its current form.